# Effect of the Dimensions of Coplanar Inner Floating Ring Electrode on the Performance of Liquid Crystal Lenses

Yung-Hsiang Hsu [iD], Bo-Yu Chen and Chia-Rong Sheu *[iD]

Department of Photonics, National Cheng Kung University, Tainan 701, Taiwan;
l78991241@gs.ncku.edu.tw (Y.-H.H.); l76061333@gs.ncku.edu.tw (B.-Y.C.)
* Correspondence: pizisheu@mail.ncku.edu.tw; Tel.:+886-6-275-7575 (ext. 63929)

**Abstract:** In this study, we mainly investigated the effect of the dimensions of a coplanar inner floating ring (CIFR) on the lens performance in CIFR–hole-patterned electrode liquid crystal lenses (HPELCLs) at 100 Hz of the electrical driving frequency. The operation and threshold voltages in CIFR–HPELCLs are approximately 76% compared with those of the conventional HPELCL. The diameter of the CIFR with 360 μm in relation to imaging capabilities and those of the conventional glass lens and HPELCL were analyzed via the modulation transfer function. The relative mechanisms of the CIFR dimensions and the lens performance were also examined. An electric circuit model was used to analyze and illustrate the experimental results.

**Keywords:** liquid crystal lens; hole-patterned electrode; floating ring electrode; coplanar electrode

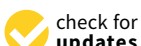

## 1. Introduction

Liquid crystals (LCs) have been studied to fabricate various electro-optical devices because of their unique physical properties, such as birefringence, response time and dielectric anisotropy [1–6]. Amongst these devices, LC lenses have been highly investigated [7–18], especially their easy fabrication processes, absence of disclination line, polarization independence and tunable focal length [19–24].

A novel structure LC lens based on a microstructured transmission line is recently proposed; it uses a transmission line acting as a voltage divider and concentric electrodes that distribute the voltage homogeneously across the active area of the LC lens [25,26]. It can modulate the LC-induced phase profile to achieve optical devices with multiple functions, such as beam steering, 2D tunable diffraction gratings and lens function. However, this structure requires high lithography accuracy, which increases the difficulty of the process fabrication of LC lens.

Hole-patterned electrode LC lenses (HPELCLs) are a type of LC lens that has been successfully designed with good lens performance, easy fabrication and simple cell structure. Thus, HPELCLs have been widely researched [27–34]. HPELCLs use an applied voltage between the hole-patterned electrode on the top substrate and the plane electrode on the bottom substrate. The intensity of the fringe field generated by the hole-patterned electrode gradually decreases from the edge to the center of the hole pattern; thus, it provides a nonuniform and circularly symmetric electric field to spatially reorient LC molecules with suitable refractive index distributions.

A suitable dielectric layer is generally required to allow electric fields to penetrate the center of an HPELCL. The generated electric fields distributed in the LC layers drive the molecular orientations to achieve ideal distributions of the gradient refractive indices with lens capabilities [35]. In addition, the dielectric layer coating on the hole-patterned electrode effectively prevents disclination line occurrence in the HPELCL that is mainly caused by the inconsistency of locally spatial LC reorientations with respect to the fringe fields. However, the inserted dielectric layer significantly increases the operation and threshold voltages of the HPELCL. This effect becomes significant as the operating frequency decreases.

Extra floating electrodes that are fabricated in HPELCLs effectively enhance the lens capabilities with a large focusing power at low operation voltages [36,37]. In the past two decades, HPELCLs have been extensively investigated and used for various electro-optical applications. In our previous work [38], the coplanar inner floating ring (CIFR)–HPELCL demonstrated a better lens performance, such as lower threshold and operation voltages at 100 Hz electric frequency, than the conventional HPELCL. This design requires only a single channel to apply an operation voltage, and the fringe field generated by the CIFR electrode can greatly reduce the operating voltage of the LC lens at low operating frequencies. Connecting the inner electrode to the external electrode/voltage with stripes can more accurately control the refractive index gradient distribution of the LC molecules; however, it requires additional channels to apply more than one operation voltage. The ion effect is obvious when the ionic impurity charge in the LC layer is caused by the DC component contained in the driving signal. The positive- and negative-charged ions will be affected by the external electric field and move to the opposite polarity electrodes in the LC layer to form electric double layers, which will affect the quality of the LC device. However, we apply AC signal operation voltage on CIFR–HPELCL, which can effectively avoid the influence of ion effect on the performance of the LC device.

In this study, we examined the improvement in lens performance with the various dimensions of CIFR electrodes based on an electric circuit model. Through equivalent electric circuit model analysis, we examined the influence of CIFR electrode dimensions on the focusing ability and imaging quality of the lens via changing the key factor $C_{glass}$ capacitance. The change in $C_{glass}$ capacitance also significantly affects the operation voltage and threshold voltage of CIFR–HPELCL, given that the fringe field intensity generated by the CIFR electrode and the cross voltage of the electric field applied to the LC layer is affected. The relationship between the capacitance $C_{glass}$ and the dimension of the CIFR electrode was also established via the commercial simulation software. The electric field distribution generated by various diameters CIFR electrode and hole-patterned electrode in the LC cell were also investigated via simulation software to compare with the results of optical experiments. The optical interference patterns of the LC lenses were obtained to calculate the maximum lens power and determine the most suitable dimension of the CIFR electrode for use as an optical lens in CIFR–HPELCL of 1 mm diameter. Modulation transfer functions (MTFs) were used to examine and compare the image resolutions of the HPELCL, the CIFR–HPELCL and the conventional glass lens.

## 2. Structure and Electric Circuit Model of CIFR–HPELCL

Figure 1 presents the cross-sectional scheme and top view of the CIFR–HPELCL with its equivalent resistor–capacitor (RC) electric circuit model. The lens consists of an LC layer and two indium tin oxide (ITO) conductive glass substrates of 1.1 mm thickness. The cell gap was controlled by Mylar spacers and filled with nematic E7 LCs (purchased from Daily Polymer, Ltd.) to achieve homogeneous cells with rubbed polyvinyl alcohol (PVA) films. The actual optically measured thickness of the LC was near 50 μm. Three CIFR ITO electrodes (*D*) with different diameters at the outside boundaries, namely, 240, 360 and 440 μm, were investigated. To prevent the occurrence of disclination lines, all the CIFR–HPELCLs were coated with an additional 82 μm thick Norland optical adhesive (NOA65, Thorlabs, Inc.) films on the hole-patterned ITO electrodes as dielectric layers [32,39]. The equivalent RC electric circuit model shown in Figure 1c was used to analyze the lens performance with the variation in the dimensions of the CIFR electrodes. The dimensions are represented by three major parts individually indicated by blue, red and yellow dotted rectangles in the figure. The dielectric constants (ε) and resistivity (ρ) of the materials at 1 kHz electric frequency are listed in Table 1 [1,40,41]. At 100 Hz electric frequency, these physical constants obtained through measurements are the same as those at 1 kHz, except for dielectric constant 6 for the NOA65 material.

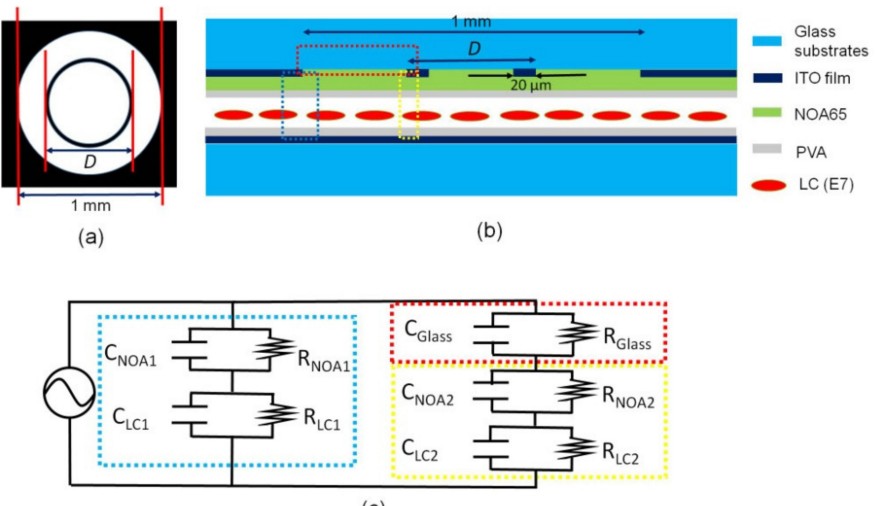

**Figure 1.** Structure and corresponding equivalent circuit model of the coplanar inner floating ring (CIFR)– hole-patterned electrode liquid crystal lenses (HPELCL). (**a**) Top view and (**b**) cross-sectional view of the proposed CIFR–HPELCL; (**c**) corresponding equivalent RC circuit composed of three parts individually indicated with colorful dotted rectangles.

**Table 1.** Dielectric constant and resistivity of the material in the CIFR–HPELCL.

| Material | Dielectric Constant $\varepsilon$ | Resistivity $\rho$ ($\Omega \cdot$m) |
|---|---|---|
| LC (E7) | 19.3 ($\varepsilon\|$), 5.2 ($\varepsilon\perp$) | $1.8 \times 10^{10}$ |
| NOA65 | 4 | $1.0 \times 10^{13}$ |
| Glass substrate | 6.9 | $1.0 \times 10^{10}$ |

Given the equivalent RC electric circuit model, the electric fields distributed in the LC layer are directly related to the voltages from the fringe fields of the hole-patterned electrode ($V_{LCH}$) and the induced electric fields of the floating ring electrode ($V_{LCF}$), which satisfy Equations (1) and (2), respectively. The symbols in both equations are defined as follows: $V$ is the driving voltage; the first grouped impedance of $Z_{LC1}$ and $Z_{NOA1}$ is related to the fringe electric field at the hole-patterned electrode; the second grouped impedance of $Z_{LC2}$ and $Z_{NOA2}$ is related to the induced electric field at the CIFR electrode, and the impedance of $Z_{Glass}$ is related to the top glass substrate between the CFIR and the hole-patterned electrodes.

$$V_{LCH} \approx [Z_{LC1}/(Z_{NOA1} + Z_{LC1})]V, \tag{1}$$

$$V_{LCF} \approx [Z_{LC2}/(Z_{Glass} + Z_{NOA2} + Z_{LC2})]V. \tag{2}$$

In general, resistance $R$ is defined as Equation (3) if the material is geometrical with $d$ thickness and $A$ area. According to the list in Table 1 for LC lens fabrications, the resistivity ($\rho$) values of NOA65 and E7 LCs are $1.0 \times 10^{13}$ and $1.8 \times 10^{10}$ $\Omega \cdot$m, respectively. Therefore, a related equation, $R_{NOA1} = 1.82 \times 10^3 R_{LC1}$, could be used.

In the equivalent circuit, the impedance of capacitor $Z_c$ is dependent on electric frequency $f$ and capacitance value $C$, which satisfies Equation (4). The total impedance ($Z$) of each dielectric layer can be expressed as the parallel connection of resistance $R$ and capacitor impedance $Z_c$, as shown in Equation (5). In the equation, $Z$ can be $Z_{LC1}$, $Z_{NOA1}$, $Z_{LC2}$, $Z_{NOA2}$ or $Z_{Glass}$.

$$R = \rho d / A, \tag{3}$$

$$Z_c = 1/(j2\pi f C), \tag{4}$$

$$Z = (R^{-1} + Z_c^{-1})^{-1}. \tag{5}$$

According to Equation (4), capacitor impedance $Z_c$ increases when the LC lens operates at low electric frequencies. Therefore, total impedance $Z$ is dominated by resistance $R$ when operating at low electric frequencies due to the parallel RC connections in each component. This condition satisfies Equation (5). Thus, the threshold and operation voltages increase in the HPELCL when operating at an electric frequency of 100 Hz instead that of 1 kHz due to that the relation $R_{NOA1} = 1.82 \times 10^3 \ R_{LC1}$ is based on Equation (1). Here, the threshold voltages corresponding to the beginning of the initial interference patterns. The operation voltages correspond to the maximum pairs of black–white concentric interference patterns in the lenses. The CIFR electrode provides an additionally induced electric field to contribute to the original fringe field from the hole-patterned electrode to enhance the final radial phase retardation when the fringe electric field is generated by the hole-patterned electrode attenuates as the electric frequency decreases [38].

According to Equation (2), the induced electric field from the CIFR electrode is related to impedance $Z_{Glass}$ in which $R_{Glass}$ and $C_{Glass}$ are related to the electric frequency ($f$). This condition satisfies Equation (6):

$$Z_{Glass} = (R_{Glass}^{-1} + j2\pi f C_{Glass})^{-1} \tag{6}$$

In general, capacitance is directly related to the geometry of the terminal electrodes, the dielectric materials and the distances between terminals. In this study, the radial dimensions of the CIFR electrodes were the main variables that affected the electrical properties of the LC cells due to the various distances between the terminal electrodes. A large $D$ of CFIR electrode achieves a large capacitance of $C_{Glass}$ due to the short distance between the CIFR and the hole-patterned terminals [39]. Thus, impedance $Z_{Glass}$ was decreased to induce an increase in $V_{LCF}$, which enhanced the fringe field generated by the CIFR electrode and effectively drove the LC molecules in turn.

Figure 2 shows the calculated $C_{Glass}$ value corresponding to the floating ring electrode diameter under the 1 mm hole-patterned electrode using the commercial software COMSOL Multiphysics (COMSOL, Inc., Burlington, MA, USA). The structure dimensions and their parameters were set as follows: The diameter of the hole-patterned electrode was 1 mm, the LC cell was 50 μm thick, the line width of the floating ring electrode was 20 μm, the NOA65 film was 82 μm thick, the PVA film was 0.1 μm thick, the ITO film was 0.1 μm thick, and the glass substrate was 0.55 mm thick. The CIFR electrode was set to 1 V via the terminal interface, and the hole-patterned electrode was set to the ground under the electrostatics module. The stationary source sweep study was used to calculate the capacitance through the integration of the surface charge density.

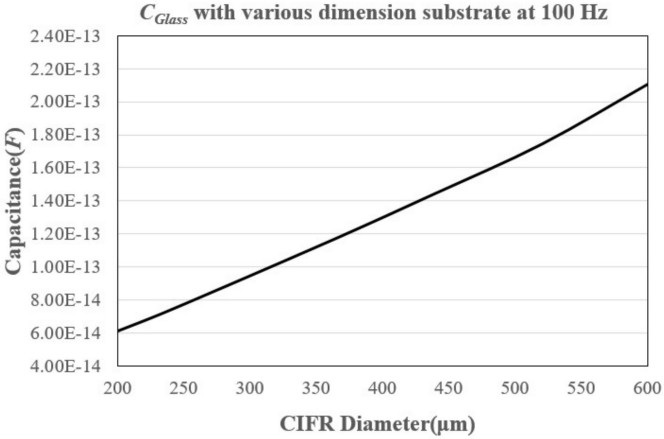

**Figure 2.** Capacitance value simulation results corresponding to the various CIFR electrode diameters under 1 mm hole-patterned electrode.

The hole-patterned and floating ring electrodes were in concentric circles. When the line width of the floating ring electrode was fixed at 20 µm, $C_{Glass}$ increased from $6.12 \times 10^{-14}$ *F* to $2.11 \times 10^{-13}$ *F* with the diameter increase of the floating ring electrode from 200 µm to 600 µm under the 1 mm hole-patterned electrode. According to Equation (2), the decrease in impedance $Z_{Glass}$ enhances the fringe field generated by the CIFR electrode and reduces the threshold and operation voltages of CIFR–HPELCLs.

Figure 3 indicates the simulation comparisons of the electric field distributions in the two layers of NOA 65 and LCs in the HPELCL and the CIFR–HPELCL with dimensions of 240, 360 and 440 µm in diameter to drive the LC molecules via COMSOL Multiphysics. In the calculation, 10 $V_{rms}$ was supplied across the hole-patterned and bottom planar electrodes, and the electric frequency was 100 Hz. The structure dimensions and their parameters were set up similarly to those in the $C_{Glass}$ simulation. The color bar shows the gradual decrease in the electric fields from red to blue.

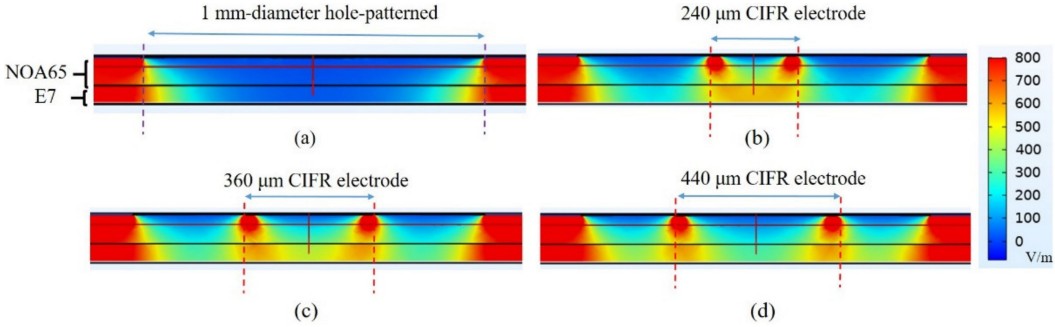

**Figure 3.** Simulation results of electric field distributions with the (**a**) HPELCLs and CIFR–HPELCLs with the diameters of (**b**) 240, (**c**) 360 and (**d**) 440 µm under the 10 $V_{rms}$ operation voltage at an electric frequency of 100 Hz. The red lines indicate the position of CIFR electrodes in the cells.

Figure 3a shows the electric field distributions of the 1 mm diameter HPELCL. The 240 µm CIFR induced a smaller fringe field than the 360 and 440 µm CIFRs due to its smaller $C_{Glass}$ capacitance, as shown in Figure 3b. By contrast, large CIFR diameters, such as 360 and 440 µm, responded to a large $C_{Glass}$ capacitance. Thus, the fringe fields induced by the CIFR electrodes achieved remarkable distributions of gradient refractive index, as shown in Figure 3c,d. The total phase retardation with respect to the distributions of the gradient refractive index in the case of the 440 µm CIFR electrode decreased in the lens center. The induced fringe electric field in the LC layer gathered around the large inner diameter (440 µm) of the CIFR electrode against the fringe field and extended to the lens center. Consequently, the relative position of the hole-patterned and CIFR electrodes achieved maximum phase retardation in the 360 µm CIFR–HPELCL due to the smooth distribution of the electric field intensity from the edge to the center of the hole-patterned electrode.

## 3. Experiment and Result Discussion

Figure 4 shows the experimental setup used to measure the interference patterns (i.e., pairs of black–white circles) of the LC lens under various electric voltages and frequencies [42]. A collimated He–Ne laser beam ($\lambda$ = 632.8 nm) with an intensity controlled by an attenuator was passed through a spatial filter and a beam expander for purification. An expanded beam with a diameter of 35 mm was then achieved and used as the final incident beam of the LC lens. Two crossed polarizers (i.e., a polarizer and an analyzer) placed in the front and rear of the LC lens, respectively, were used to record the interference patterns in the LC lens. Both polarization directions of the polarizers were at a 45° angle to the rubbing directions of the LC alignment in the cell. A waveform generator (33220A, Agilent) was used to provide an AC electrical signal (i.e., a square waveform with 100 Hz frequency) that was amplified 20 times via a voltage amplifier (FLC Electronics AB, Broadband Linear Amplifier F20A) to connect the LC lens. In the end, a charge-coupled device (CCD) camera (DVC Company, 1500 M-00-CL) was used to record the interference pattern. The various

AC voltages were connected to two terminals, namely, hole-patterned and the entire ITO electrodes, in the CIFR–HPELCL and the HPELCL.

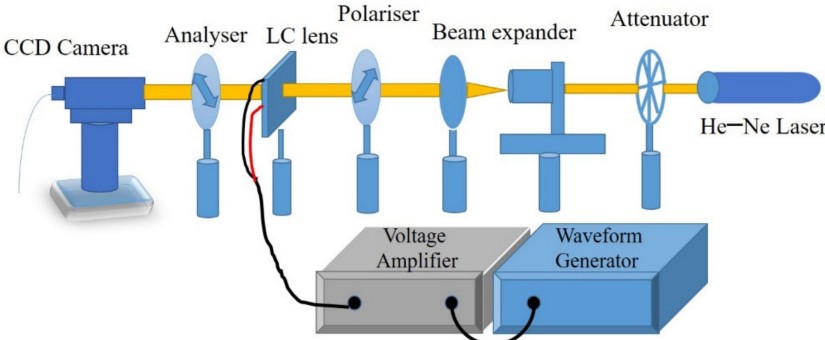

**Figure 4.** Experimental setup for measuring and evaluating the electro-optical performance of the fabricated liquid crystal (LC) lens.

Figure 5 shows the maximum pairs of concentric black–white interference patterns in the HPELCL and the CIFR–HPELCL with *D* radial dimensions operating at an electric frequency of 100 Hz with square waveforms. A pair of black–white concentric circular interference patterns indicates the $2\pi$ phase retardation in the radial direction. The total number of pairs is denoted as *N*. R in red is the rubbing direction, and A and P in black are the directions of the analyzer and the polarizer, respectively. The patterns resulted in pair numbers *N* = 12, 11, 14, 10 that correspond to the HPELCL and the CIFR–HPELCLs with CIFR electrode diameters of 240, 360 and 440 μm. Although the operation voltages of the HPELCL and the CIFR–HPELCLs increased due to the low electrical driving frequency (100 Hz), the operation voltages of the CIFR–HPELCLs were 12, 12 and 8 $V_{rms}$, which correspond to CIFR electrode diameters of 240, 360 and 440 μm, respectively. These values are lower than the operation voltage increase to 50 $V_{rms}$ in the HPELCL.

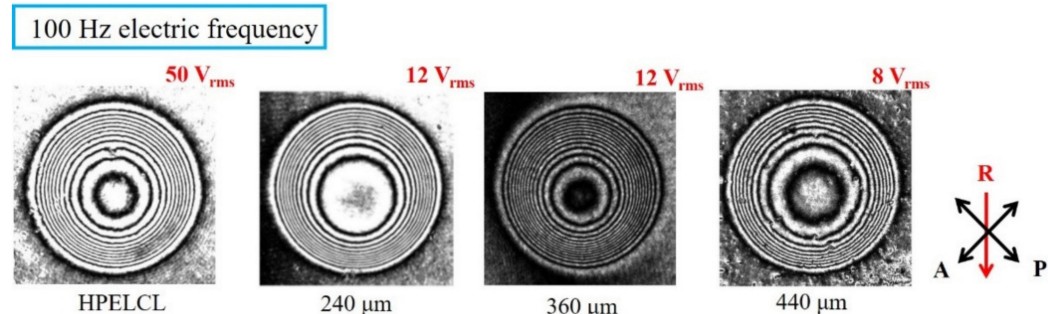

**Figure 5.** Optical comparisons of the black–white concentric interference patterns with respect to individual operation voltages at the electric frequency of 100 Hz in the HPELCL and the CIFR–HPELCLs with the radial dimensions of the CIFR electrodes.

Table 2 lists the threshold and operation voltages with the maximum pairs of black–white concentric interference patterns (*N*) at an electric frequency of 100 Hz in the HPELCL and the CIFR–HPELCLs with radial dimensions of the CIFR electrodes. The HPELCL required larger threshold and operation voltages than the CIFR–HPELCL at the electric frequency of 100 Hz because low electric frequency *f* resulted in large capacitance impedance $Z_{NOA1}$ and $Z_{LC1}$, which decreased $V_{LCH}$. This condition satisfies Equation (1). The threshold voltages also decreased simultaneously. The large dimensions of the CIFR electrodes achieved low threshold voltages. This trend is consistent with Equation (2) due to the decrease in impedance $Z_{Glass}$. The threshold voltages of the CIFR–HPELCLs were related to the diameter of the CIFR electrode under the electric frequency of 100 Hz. The threshold

voltage decreased under a 440 μm CIFR electrode compared with those under the 240 and 360 μm CIFR electrodes due to the increased fringe field generated by the CIFR $V_{LCF}$.

**Table 2.** Comparison of various diameters of floating ring electrodes.

| CIFR Electrode Diameter | Threshold Voltage | Operation Voltages with Maximum Pairs (Voltages/*N*) |
|:---:|:---:|:---:|
| No CIFR | 13 $V_{rms}$ | 50 $V_{rms}$/12 |
| 240 μm | 4 $V_{rms}$ | 12 $V_{rms}$/11 |
| 360 μm | 3 $V_{rms}$ | 12 $V_{rms}$/14 |
| 440 μm | 2 $V_{rms}$ | 8 $V_{rms}$/10 |

However, the distributions of the gradient refractive index were due to combining the spatially electric fields with the fringe fields at the hole-patterned electrode and the induced electric fields at the CIFR electrode. Therefore, the CIFR–HPELCL with a 360 μm CIFR electrode obtained the maximum pair number *N* = 14 at 12 $V_{rms}$. This value is larger than those of the CIFR–HPELCLs with 240 and 440 μm diameter CIFR electrodes (i.e., *N* = 11 at 12 $V_{rms}$ and *N* = 10 at 8 $V_{rms}$, respectively).

Figure 6 presents the curve fitting of the radial phase retardation in the CIFR–HPELCLs with radial dimensions of CIFR electrodes with respect to the operating voltages at 100 Hz electric frequency. By contrast, the phase distribution profile of the 360 μm CIFR electrode was deeper into the center of the lens than those of the 240 and 440 μm CIFR electrodes. This finding can be attributed to the electric field gradient distribution's maximization of the refractive index gradient distribution of the LC molecules.

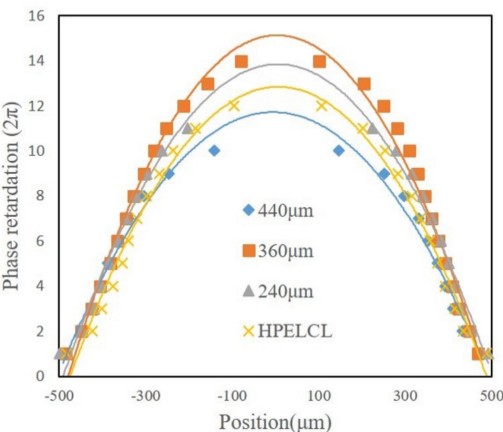

**Figure 6.** Radial phase retardation and curve fitting of the HPELCL and the CIFR–HPELCLs with the individual CIFR electrodes of different diameters (i.e., 240, 360 and 440 μm) under the electric frequency of 100 Hz with operation voltages.

Table 3 shows a comparison of the wavefront error root-mean-square (RMS) values of the HPELCL and the CIFR–HPELCL with 240, 360 and 440 μm CIFR electrodes under an electric frequency of 100 Hz. The RMS of the wavefront error is generally used to evaluate the lens performance compared with an ideal parabolic refractive index profile. An RMS value smaller than 0.07 $\lambda$ is considered acceptable for lens performance for imaging applications [43]. The RMS values in the fabricated HPELCL and CIFR–HPELCL with 240, 360 and 440 μm CIFR electrodes under an electric frequency of 100 Hz were 0.59, 0.54, 0.56 and 0.52 $\lambda$. The CIFR–HPELCL presented a better lens performance in the imaging applications than the HPELCL under the electric frequency of 100 Hz.

**Table 3.** Comparison of the wavefront error root-mean-square values of the HPELCL and CIFR–HPELCL with radial diameter CIFR electrodes at an electric frequency of 100 Hz.

| HPELCL | 240 µm CIFR Electrode | 360 µm CIFR Electrode | 440 µm CIFR Electrode |
|---|---|---|---|
| 0.59 $\lambda$ | 0.54 $\lambda$ | 0.56 $\lambda$ | 0.52 $\lambda$ |

According to Equation (7), the focal length of the HPELCL and the CIFR–HPELCLs is related to the number of interference fringes with the appropriate spatial distribution [44]:

$$f_{lens} = \frac{r^2}{2\lambda N},\tag{7}$$

where $r$ is the radius of the hole-patterned electrode, $N$ represents the pairs of black–white concentric interference patterns of the LC lens, $\lambda$ is the wavelength of the incident light, and $f_{lens}$ denotes the focal lengths of the HPELCL and the CIFR–HPELCLs under the applied voltages. Figure 7 shows the tuneable focuses at the electric frequency of 100 Hz with the HPELCL and the CIFR–HPELCLs with respect to the applied voltages in the LC lens.

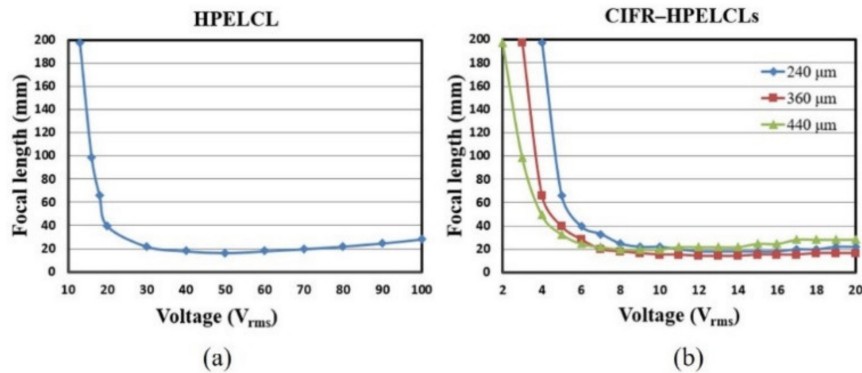

**Figure 7.** Electrically tunable focal lengths of the (**a**) HPELCL and (**b**) the CIFR–HPELCLs with individual CIFR electrodes of different diameters (i.e., 240, 360 and 440 µm) under the electric frequency of 100 Hz.

As shown in Figure 7a, the HPELCL shows the minimum foci, that is, 16.45 mm, at an operating voltage of 50 V_{rms}. The fabricated CIFR–HPELCLs also possessed the minimum foci, namely, 17.95, 14.10 and 19.74 mm in the 240, 360 and 440 µm CIFR electrodes at 12, 12 and 8 V_{rms} operation voltages, respectively. These findings can be attributed to the embedded CIFR electrode that enhanced the fringe field and reduced the operation voltage, as shown in Figure 7b.

The imaging capabilities of the HPELCL and the CIFR–HPELCLs under an electric frequency of 100 Hz was evaluated as follows: The fabricated CIFR–HPELCL with a 360 µm CIFR electrode was used for image analysis because it has the shortest minimum foci. A 1951 USAF (U.S. Air Force) resolution test chart was used as the object and placed in front of the LC lens to analyze the image resolution, as shown in Figure 8a. The red circle indicates the target used to analyze the lens imaging. It is located in the group 1 target 3 straight pattern of the 1951 USFA, and the line width is 198 µm.

The distance between the LC lens and the object was 65 mm, and a polarizer was placed between the LC lens and the object with a transmission axis parallel to the rubbing direction of the LC cell. The images were recorded using a CCD camera (DVC Company, 1500M-00-CL), which was placed behind the LC lens. Figure 8b presents the straight pattern image of a part of the 1951 USAF resolution test chart recorded by the CCD without an LC lens. Then, the LC lens was placed in front of the object and operated at the minimum foci under 100 Hz electric frequency. Figure 8c,d shows the straight pattern recorded by the CCD camera via the HPELCL at 50 V_{rms} and the CIFR–HPELCL with a 360 µm CIFR

electrode at 12 $V_{rms}$, respectively. The image captured via a conventional glass lens with a 30 mm focal length as a reference is shown in Figure 8e.

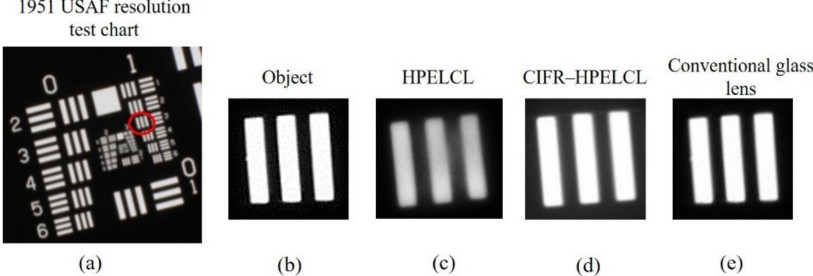

**Figure 8.** Imaging evaluation of the proposed CIFR–HPELCL. (**a**) 1951 USAF resolution test chart. (**b**) Straight pattern image recorded via the charge-coupled device (CCD) without a lens. Straight pattern image with the (**c**) HPELCL at 50 $V_{rms}$, the (**d**) 360 μm CIFR electrode CIFR–HPELCL at 12 $V_{rms}$ under the 100 Hz electric frequency and the (**e**) conventional glass lens.

The commercially available software Quick MTF was used to calculate the MTF using Equation (8).

$$MTF = \frac{I_{max} - I_{min}}{I_{max} + I_{min}}, \tag{8}$$

where $I_{max}$ and $I_{min}$ denote the maximum and minimum intensities in the image, respectively, in general, the MTF defines the ability of an optical system to resolve a contrast at a given spatial frequency and falls from 1 to 0 as the spatial frequency increases. Figure 9 shows the MTF curve calculated from the images in Figure 8b,c,d. At spatial frequencies from 0.3 to 0.4 cycles/pixel, the MTF curve of the HPELCL descended to 0, similar to that of the CIFR–HPELCL. However, the MTF curve of the HPELCL descended rapidly below the spatial frequency at 0.1 cycles/pixel compared with that of the CIFR–HPELCL. Therefore, the fabricated CIFR–HPELCL with a 360 μm CIFR electrode at 12 $V_{rms}$ operation voltage exhibited better imaging performance and image resolutions than the HPELCL at 50 $V_{rms}$ operation voltage under the 100 Hz electric frequency. Once its dimension matches the diameter of its hole-patterned electrode, the fringe field achieves the maximum refractive index gradient distribution of LC molecules in the active area of the LC lens. The CIFR electrode will not damage the imaging quality of the lens because the CIFR electrode has a line width of only 20 μm and is a transparent ITO electrode. Thus, it will not impair the lensing performance of the device. By contrast, the conventional glass lens exhibited the best performance of the MTF curve, which decayed slowly with the increasing spatial frequency.

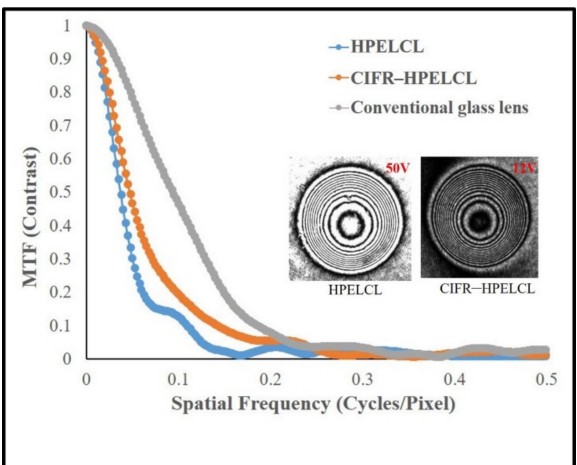

**Figure 9.** Modulation transfer functions (MTFs) of the fabricated HPELCL, the CIFR–HPELCL with a 360 μm CIFR electrode under the 100 Hz electric frequency and the conventional glass lens.

## 4. Conclusions

The proposed CIFR–HPELCL with suitable CIFR electrode dimensions achieved a lens performance with large-phase retardation penetrating the central area of the lenses and low operation voltages at the electric frequency of 100 Hz. The fringe field intensity of the CIFR electrode was mainly affected by the capacitance ($C_{Glass}$) between the CIFR and hole-patterned electrodes. The fringe field generated by the CIFR electrode increased with the increase in $C_{Glass}$. However, with the increase in the CIFR electrode dimension, the fringe field generated by the inner CIFR electrode cannot enter the central working area of the LC lens due to the relative position geometry of the hole-patterned, CIFR and planar ITO electrodes. Consequently, the 360 μm CIFR electrode achieved the maximum phase retardation under the CIFR–HPELCL of 1 mm diameter in this study.

The proposed CIFR–HPELCL with an aperture dimension of 1 mm and a 360 μm CIFR electrode was electrically operated at the operating voltage of 12 $V_{rms}$ and the electric frequency of 100 Hz to achieve the maximum radial phase retardation, that is, $N = 14$. The operation voltage (from 50 $V_{rms}$ to 12 $V_{rms}$) and threshold voltage (from 13 $V_{rms}$ to 3 $V_{rms}$) of the 360 μm CIFR–HPELCL successfully decreased by approximately 76% for the electric operation in the HPELCL at the electric frequency of 100 Hz. The imaging capabilities of the proposed CIFR–HPELCL, the HPELCL and the conventional glass lens via the MTF curves were compared. The influence of the dimensions of each layer and the dielectric constants of each material on the threshold and operation voltage trends of the CIFR-HPELCL can be predicted via the proposed equivalent circuit model.

**Author Contributions:** Y.-H.H. and C.-R.S. conceived the experiments. Y.-H.H. and B.-Y.C. performed the experiments. Y.-H.H. analyzed the data and wrote the paper. C.-R.S. supervised the study. All authors have read and agreed to the published version of the manuscript.

**Funding:** This research was funded by the Ministry of Science and Technology, Taiwan: MOST 107-2221- E-006-149.

**Conflicts of Interest:** The authors declare no conflict of interest.

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
