# Peer review of "Effect of the Dimensions of Coplanar Inner Floating Ring Electrode on the Performance of Liquid Crystal Lenses"

_crystals, doi:10.3390/cryst11020200_

Round 1

Reviewer 1 Report

The manuscript theoretically and experimentally investigates the effect of including a floating potential electrode (FPE) in a standard hole-patterned liquid-crystal lens (HPLCL). It is demonstrated that the inclusion of the FPE can reduce the threshold and operation voltages of the device and provide a degree of freedom to tailor its optical properties.

The results are quite clearly presented and interesting. The proposed method provides an extra means of adjusting the performance of HPLCL. However, there are several points that need to be clarified before publication of the manuscript can be considered:

[1] The authors have already presented the concept of FPE-HPLCL in Ref. 30 as they acknowledge in the Introduction. It is not clear what are the novel aspects investigated in this work since all major results have been presented in Ref. 30. Please stress more the novelty of the results presented in the manuscript.

[2] The FPE essentially reduces the effective hole aperture of the lens by creating an inner electrode with floating potential that creates a fringe-field of its own. What are the main advantages of this approach? Could the lens performance be comparable by simply reducing the aperture of a standard HPLCL? The FPE creates some complicated electric field profile, as in Fig. 3, and hence fringe-field pattern. Doesn’t this impair the lensing performance of the device? Also, keeping the inner electrode at floating potential somehow complicates the design as the electrostatic problem has to be solved each time in an inverse-design approach. Would connecting with some stripes the inner electrode to the external electrode/voltage facilitate the design or not?

[3] It would be better to show a 3D schematic of the lens structure or at least a 2D cross-section of the electrode plane for those readers that are not so familiar with LC-lens designs.

[4] Table 1 and text: the dielectric constant is a relative value, it should not have units. Also, references are missing for the values reported in Table 1.

[5] Please elaborate more on how Cglass was calculated in Fig. 2. Which COMSOL model was used? Was the capacitance calculated by integrating surface charges?

[6] Figure 3: a) it is hard to distinguish the colorbars. b) what are the floating potential values for the three cases studied? c) was the fully anisotropic LC dielectric permittivity tensor properly calculated and coupled with the electrostatic problem?

[7] Was Fig. 6 calculated for the operation voltages as in Table 2 for each case investigated? Please clarify.

[8] An alternative technique was recently proposed for LC lenses with graded-index profiles based on the use of electrode stubs connected to a strip electrode that provides the desired voltage profile (dois: 10.1038/s41598-020-67141-z and 10.1038/s41598-020-70783-8). The authors are invited to briefly discuss it in the Introduction and compare with the proposed technique.

Author Response

Please see the author reply letter.

Thanks.

Reviewer 2 Report

In this manuscript, the authors developed an electric circuit model to optimize the coplanar floating ring electrode dimension for improving the performance of hole-patterned liquid crystal lens. Detailed procedures are described and confirming experiments are conducted to validate the model. It is an interesting paper, and I recommend it for publication in Crystals after following specific questions and comments have been addressed:

  1. The employed E7 liquid crystal layer is 50-mm thick. What is the response time of the LC lens? Based on the results published in Appl. Opt. 26, 3441 (1987), the visco-elastic constant of E7 at the room temperature is 20 ms/mm2. Therefore, the estimated response time (free relaxation time) is about 5s. Does this value agree with your measured result? Of course, the gray-to-gray transition time depends on the applied voltage, as derived in Appl. Opt. 28, 48 (1989). It could be faster or slower than the free decay time, depending on the gray levels. Several approaches to expedite the lens response time have been reviewed by S. Xu, et al. Micromachines 5, 300 (2014). This reference will help readers to understand how to improve the response time.
  2. In Table 1, why is the resistivity of glass substrate so low? Is this ITO-glass or bare glass?
  3. In Fig. 7: Why does the focal length bounce back as the applied voltage keeps increasing?
  4. In Fig. 8: It would be nicer to show the air force resolution target so that the readers can compare the resolution of the lenses. Images (b) and (c) seem to have blurs.
  5. Introduction: At the end of first sentence, citing a book or review paper will help readers to better understand this rich field. Liquid crystals are not only dominating display devices but also have other useful applications.
  6. The English of this manuscript should be improved.

Author Response

(The authors gave the same response as above.)

Round 2

Reviewer 1 Report

The authors have provided a satisfactory response to the questions raised by the reviewers. The manuscript is recommended for publication.